# Maternal Decisions on Portion Size and Portion Control Strategies for Snacks in Preschool Children

**DOI:** 10.3390/nu11123009

**Published:** 2019-12-09

**Authors:** Sophie Reale, Colette Marr, Joanne E. Cecil, Marion M. Hetherington, Samantha J. Caton

**Affiliations:** 1School of Health and Related Research (ScHARR), University of Sheffield, Sheffield S1 4DA, UK; s.reale@shu.ac.uk (S.R.); c.kearney@sheffield.ac.uk (C.M.); 2Population and Behavioural Sciences, School of Medicine, University of St Andrews, St Andrews KY16 9TF, UK; jc100@st-andrews.ac.uk; 3School of Psychology, University of Leeds, Leeds LS2 9JT, UK; m.hetherington@leeds.ac.uk

**Keywords:** portion size, preschool children, mothers, snacks, qualitative, think-aloud method

## Abstract

Caregivers are responsible for the type and amount of food young children are served. However, it remains unclear what considerations caregivers make when serving snacks to children. The aim of the study was to explore mothers’ decisions and portion control strategies during snack preparation in the home environment. Forty mothers of children aged 24–48 months participated in the study. Mothers prepared five snack foods for themselves and their child whilst verbalizing their actions and thoughts. Mothers were then asked about their portion size decisions in a semi-structured interview. Transcripts were imported into NVivo and analyzed thematically. Three key themes were identified: (1) portion size considerations, (2) portion control methods, and (3) awareness and use of portion size recommendations. Transient, food-related situational influences influenced mothers and disrupted planning and portion control. Food packaging and dishware size were used as visual cues for portion control; however, these vary widely in their size, thus emphasizing mothers’ uncertainty regarding appropriate portion sizes. Mothers called for portion size information to be accessible, child-centered, and simple. These findings reveal multiple considerations when deciding on the correct snack portion sizes for children. These decisions are complex and vary across situations and time, and according to the types of snacks offered.

## 1. Introduction

Young children consume larger quantities of food in the presence of larger food portion sizes, which may be attributable to social norms, visual cues, or changes in the microstructure of eating [1]. This is known as the portion size effect (PSE). This effect is robust and reliable and was recently demonstrated to persist over five days without compensatory behaviors in children aged 3–5 years [2]. Large portion sizes of energy-dense foods are becoming more easily accessible and available in the obesogenic environment, such that manufacturers are typically packaging snacks in portion sizes up to and beyond 2.5 times larger than necessary for young children [3]. Furthermore, portion sizes of food consumed within the home are also increasing [4,5]. The impact of these ongoing changes to children’s dietary intake were recorded [6,7] and suggested that children aged 1–4 years are being offered high-energy-density snacks (HED > 2.5 kcal/g as defined by reference [8]) up to three times the recommended weekly amount at one eating occasion. Frequent consumption of HED snacks is associated with a higher body mass index (BMI) in adolescents and an increased risk of associated disease such as type 2 diabetes [9]. This highlights the need to explore what influences snack portioning practices made by caregivers.

Mothers have unique views and experiences of feeding their children. For example, qualitative research suggests that there is a degree of variation in the influences and strategies used to control the portion sizes of food items served to young children [10]. Very few parents regularly use measurements or expert recommendations. Instead, caregivers self-report relying on contextual factors such as time of day, proximity to next eating occasion, and perceived or reported child hunger to determine an appropriate portion size to serve [10,11,12]. Furthermore, some parents place restrictions on the portion sizes that they offer to their children based on the perceived healthiness of the food item [10,13]. For example, in semi-structured interviews with low-income mothers from the United States of America (USA), it was revealed that some parents allow their children to consume *ad libitum* quantities of foods that they perceive to be healthy (e.g., fruits and vegetables), whereas items they perceive to be unhealthy (e.g., HED snacks) are more likely to be restricted [13]. In contrast, mothers from the United Kingdom (UK) discussed providing liked foods in large portion sizes in response to their child’s request as a means to influence their child’s behavior or to prevent upset [14].

Several portion control strategies were identified from parents from the USA, including the use of bowl sizes, small containers, and hand size. However, in the UK, there is limited research exploring caregiver’s portion control methods. Furthermore, over half of the sample in a recent study was unable to articulate how they determined an appropriate child snack portion size to serve, suggesting that portioning practices may be somewhat automatic rather than a carefully deliberative process [13]. Johnson et al. [10] used the think-aloud method, and invited mothers to describe their actions, feelings, and decisional processes during preparation of an evening meal for their child. This method was found to stimulate thoughts, reduce bias, and allow for a more thoughtful consideration of feeding behaviors that mothers may not typically verbalize. The think-aloud method also identified associations between maternal and child portion size at an evening meal, such that child portion size was positively related to maternal portion size [15]. This could be affected by parental hunger, food liking, or confusion regarding what constitutes a child-friendly portion size (e.g., reference [16]). However, not much is known about how caregivers determine a suitable portion size to offer to young children, particularly around snacking [13], and especially in the UK. Therefore, the primary aim was to explore what factors influence mothers’ decisions and judgements about a suitable snack portion size to serve preschool children and to further explore what portion control methods mothers adopt in the home environment, using the think-aloud method and semi-structured interviews. The secondary aim was to compare the snack portion sizes mothers served in the home environment to evidence-based recommended amounts [17].

## 2. Materials and Methods

### 2.1. Participants and Recruitment

Mothers (*n* = 40) of children aged 24 to 48 months old were recruited to take part in a home-based study. Half of the sample were recruited via university emailing lists, social media advertisements (e.g., Facebook, Twitter), and within toddler groups, whilst the remaining 20 participants (from Sheffield (UK) and surrounding areas) were recruited on completion of a previous online study (Reale et al., in preparation). Inclusion criteria were as follows: caregivers who were ≥ 18 years old and responsible for the food their child consumed in the home environment. Exclusion criteria included mothers and children with food allergies. All participants provided informed consent for inclusion before they participated in the study. The study was reviewed and approved by the School of Health and Related Research Ethics committee at the University of Sheffield (#011913). Mothers were compensated for their time with a £10 gift card (which can be redeemed in over 20,000 stores, restaurants, and attractions) and provided with all the snack items required for study participation.

### 2.2. Design

The study was carried out in the participants’ home and took place 2.5 h following lunch. This was carried out to ensure ecological validity for a typical snack time. The qualitative component included a think-aloud task and a semi-structured interview to explore decisions surrounding food portion sizes. The think-aloud method produces more reliable data from participants than conducting interviews alone [18], and the combined method was found to stimulate thoughts [19] and unveil feeding behaviors that are rarely verbalized [10]. The quantitative component included an objective measurement of snack portion size (g) served by participants for themselves and their child; these were compared to More and Emmett’s [17] proposed portion sizes for preschool children and manufacturer information based on a 2000 kcal/d diet [20] for adults.

### 2.3. Procedure

Mothers were instructed to serve themselves and their child a sandwich of their choice for lunch at midday and then not consume anything until the researcher arrived 2.5 h later, to attempt to standardize hunger levels across participants.

Upon arrival at the participants’ home, the time of the previous meal was verified by the researcher. Next, mothers were provided with five snack items (Table 1) one at a time by the researcher, and they were invited to verbalize their actions and thoughts whilst preparing and plating each snack item for their child as they normally would. For example, *“I’d like you to show me how you prepare your child’s snack. I want you to imagine that your child has asked for a snack, they are hungry, having not eaten for two and a half hours following lunch. This is where we will use the think-aloud method. I would like you to explain what you are doing and what you are thinking about.”* Mothers were also asked to prepare each snack for themselves, and the order was counterbalanced. To reduce priming effects and emphasize that mothers should consider how much of each individual snack they would serve in isolation, rather than collectively, all snacks remained out of participants’ view in an opaque bag. Once each snack was served onto the plate or bowl, it was immediately placed into a pre-labeled opaque bag and removed from view.

Following the think-aloud part of the study, a semi-structured interview was carried out with mothers to elicit further information [21], to prompt description of underlying decisions and motivations for serving the chosen portion sizes. The think-aloud task and semi-structured interviews were audio-recorded. At all times, the prepared snacks and plate/bowl were in view for reference. The home visit lasted approximately one hour, with each task (think-aloud task, interview, and weighing task) taking around 20 min each. On completion of the interview, the child’s height (m) and weight (kg) were measured (see Section 2.4.4), and mothers’ height and weight were self-reported. Furthermore, snacks were weighed to the nearest gram (Salter Essentials bowl scale). The researcher completed field notes before, during, and after the home visit.

### 2.4. Materials and Measures

#### 2.4.1. Development of the Interview Guide

Sample questions relevant to the research questions were devised (Appendix A). The questions were mainly open-ended to assist participants in providing detailed responses. Prompts and follow-up questions were developed to elicit more detail where necessary. The interview guide was edited during pilot interviews, in alignment with Bryman’s development of a finalized interview guide [22]. A few questions were re-worded for clarity, and an additional question, regarding purchasing smaller packaged snacks, was added. The final interview guide consisted of 20 open-ended questions.

#### 2.4.2. Snack Foods

Evidence-based portion size ranges for preschool children were proposed, by combining published data from two national surveys (National Diet and Nutrition Survey [23] and Avon Longitudinal Study of Parents and Children [24]) [17]. Foods were categorized into five food groups (1. bread, rice, and potatoes; 2. fruit and vegetables; 3. milk, yoghurt, and cheese; 4. meat, fish, eggs, nuts, and pulses; and 5. foods high in fat and/or sugar) and two food groups (groups 2 and 5) were split further to provide flexibility in serving frequencies and to reflect snack foods. One snack item from each snack food group, as defined by More and Emmett [17], was selected to ensure the inclusion of sweet and savory, unit and amorphous, and HED and low-energy-density (LED) snacks (low, ≤ 2.5 kcal/g; high, > 2.5 kcal/g) [8]. The selected snack items (Table 1) were identified as being familiar and regularly consumed by children [25] and adults [8,25], readily available in supermarkets; they would not get damaged during transportation to the study site and would not require immediate consumption after being placed into a food bag, to prevent food spoilage and waste.

Recommended snack portion sizes were informed by suggestions from More and Emmett [17] for children, whilst recommended adult snack portion sizes were provided in line with manufacturer information based on a 2000 kcal/day diet [20]. These were used as the comparator against self-selected child and adult portion sizes in order to explore the accuracy of snack portion size selection.

#### 2.4.3. Portion Size Selection

Each snack food item was removed from its original packaging, pre-weighed to the nearest gram, and placed into an opaque zip lock bag. Snack items were presented in quantities four times the recommended amount to prevent a ceiling effect. All snack items served were also weighed to the nearest gram, as a measure of portion size.

#### 2.4.4. Anthropometrics

Each child’s height (cm) (Leicester height measure: child growth foundation) and weight (kg) (Marsden M-420W portable floor scale) were measured by the researcher. Weight-for-height *z*-scores were calculated using the World Health Organization anthropometric calculator (http://www.who.int/childgrowth/software/en/).

### 2.5. Data Analysis

The qualitative data (think-aloud task and semi-structured interview) were combined as demonstrated previously by Johnson et al., [10], and transcribed verbatim. Transcripts were imported into NVivo for thematic analysis (SR). Thematic analysis was chosen as it emphasizes, records, and examines patterns within the data following six phases to reveal how each theme is related to the narrative as a whole [26]. The analysis began with data familiarization; transcripts were read and then re-read to achieve immersion in the data and to begin identifying possible patterns [27]. Initial codes were formed by clustering patterns in words and phrases, and the data were coded inclusively (i.e., text before and after the section of interest was coded) to maintain context throughout the analysis [28]. Codes were then grouped into themes using an inductive approach to ensure themes were connected to the data rather than the researcher’s interest in the topic [26]. Subthemes were also formed to provide structure and demonstrate hierarchy within themes. The entire dataset was re-read to check the validity of individual themes in relation to the dataset and whether the thematic map reflected the cohort’s responses to interview questions. A total of 10% of manuscripts were independently coded by a second reviewer (C.M.), and key themes were agreed upon according to their prevalence within the dataset and importance to the research question [26]. In the final phase, a rich thematic description of the entire dataset was produced to provide a deeper understanding of participant’s actions. Data extracts were chosen based on their relevance to the area of interest and were embedded within an analytical narrative to describe and support the outcome of the research question.

Quantitative data were entered into SPSS (IBM SPSS Statistics v22, Armonk, NY, USA). Snack portion sizes served (g) for caregivers and children are presented as means (± SD) and ranges. Amounts served were compared to suggested portion sizes for children [17] and adults (manufacturers’ recommendations based on a 2000 kcal/day diet) for each food using an independent sample *t*-test. Furthermore, a Pearson correlation was conducted to explore the relationship between the snack portion size mothers served themselves and their child. Significance was established at *p* < 0.05.

## 3. Results

### 3.1. Participant Demographics

A total of 40 mother–child dyads completed the home-based study. Mothers had a mean age of 35.0 ± 4.5 years. Most mothers were educated to at least high-school level (95% ≥ A-level or equivalent), employed (85%), white British (95%), and on average just outside healthy weight range (mean (M) = 25.5 ± 5.4 kg/m^2^). According to the index of multiple deprivation, caregivers were from diverse socioeconomic backgrounds (40% residing in areas of the UK which fall below the median decile for deprivation) [29], with almost one-quarter of caregivers earning below the average household income for 2017 [30]. Children had a mean age of 34.7 ± 8.6 months (62% male) and, on average, were of a healthy weight (BMI centile = 82.2 ± 17.3, *z*-score = 1.2 ± 0.8).

### 3.2. Qualitative Results

#### 3.2.1. Theme 1: Portion Size Considerations

Detailed discussions regarding the factors that influence mothers’ portion size decisions when serving themselves and their children snack foods occurred during the think-aloud task and interviews. Several situational factors were revealed which were categorized into four subthemes (Table 2). Situational factors included features of the environment (including proximity to the next or last meal occasion), attributes of the mother herself (including what she was served as a child), features of the foods (including how much that food is liked by the child), and perceived child hunger and food preferences.

##### Features of the Environment

When deciding upon how much to serve, mothers discussed how this varied significantly throughout the day based upon their child’s intake so far and proximity to the next meal occasion. When the snack offering was close to a meal, a smaller portion would be provided. Alternatively, if the child missed a meal or had to wait a significant amount of time before the next eating occasion, then a larger portion would be served (*“Like today she didn’t eat much at lunchtime so I probably would tend to give her a bigger snack” P19, daughter, 43 months*).

Factors within the immediate environment of the snack offering, such as food availability, appeared to be largely influential in mothers’ decisional processes when determining an appropriate snack portion size to serve (*“Sometimes I find, oh there’s just three left so I’m like oh, I may as well dish them out” P7, daughter, 42 months*). When limited quantities of food remained, a smaller portion would be provided.

Other factors that vary throughout the day, such as children’s activity levels, behavior, and hunger were also discussed in detail. Mothers felt their child required more food if they had been, or were about to be, physically active.

##### Maternal Hunger and Expected Intake

Mothers were aware that sometimes their own hunger, food liking, and portion size influenced the snack portion size they would serve to their child (*“I suppose it is often based on how much I think I might eat” P14, son, 47 months*). For example, when hungry, a mother was more likely to provide their child a larger snack portion size as they assumed their child must also be hungry. Many mothers discussed having a desired amount they wanted their child to consume (*“I do have it in my head that I want her to have had a certain amount in the day” P3, daughter, 45 months*). This was often not based on recommendations but merely a quantity that they felt suitable for their child, possibly based on past consumption experiences. Furthermore, mothers reported relying on their own experience as a child and observing other mothers’ feeding practices, when serving their child snack foods (*“More often I think I just judge based on maybe what my parents would have given me as a child or what I see other children having” P8, son, 29 months*).

##### Features of the Food

Nutritional content (sugar, salt, and fat) and perceived healthiness of snack items appeared to influence the portion size mothers serve. For example, food items that are perceived to be healthy were served in larger portions or ad libitum, to encourage healthy consumption (*“Generally if it’s healthy I’ll give her lots and lots. If its healthy stuff she can have as much as she likes” P18, daughter, 28 months*). Alternatively, foods containing larger quantities of sugar and salt were served in smaller portions to encourage healthy consumption and good dental health.

Mothers also considered ease of consumption and messiness of the food when deciding how much to provide. Foods such as raw carrot, which may be difficult and, thus, take more time for a child to eat, were served in smaller portion sizes. Similarly, food that creates mess during consumption, such as chocolate cookies, was served in smaller portions.

##### Child Trait and State Food Preference and Hunger

Child food liking and the amount mothers believed their child could consume appeared to influence the amount mothers were willing to serve to their child. A selection of mothers felt that their child would always eat the entire snack that was offered to them (*“I tend to give him snacks he likes; I expect he would eat all of it. I think snacks aren’t something that you leave” P17, son, 41 months*). However, other mothers felt that their child may leave a small amount, especially if it was a novel or less liked item. Therefore, mothers provided small portion sizes or none at all, to prevent food waste, when a less-liked food was on offer (*“If I give him a snack and he doesn’t like it, I’m probably not going to give it to him again” P17, son, 41 months*).

Similarly, mothers expressed relying upon interpretations of their child’s momentary hunger and appetite to guide their decision regarding portion size, and they did not believe their child could over-eat (*“It depends how hungry she is. If she is hungry she’s going to eat. If she’s not hungry then she’s not going to eat” P10, daughter, 30 months*).

In some cases, mothers would provide their child with the quantity their child requested and, thus, allowed their child to directly guide their decisions. In other cases, mothers identified a suitable portion size based on how much their child usually eats (*“I think just experience really because I’ve been putting things in her sandwich box most nights and I just kind of know what she is going to eat” P11, daughter, 39 months*). Alternatively, mothers expressed providing larger food portion sizes of HED snacks to control behavior (*“If he finished off the crisps and wanted more, and it was going to lead to upset I’d definitely give him more” P20, son, 29 months*).

#### 3.2.2. Theme 2: Methods Used to Control Portion Sizes Served

Mothers discussed and also demonstrated a variety of methods to control the portion size that they offered to their child including package/unit or dishware size, subdividing larger portions into small portion sizes, sharing snacks between multiple children, offering an initial small portion size in anticipation that the child would request more, or breaking units into multiple smaller items to create an illusion that more is being offered. These portion control methods were categorized into four subthemes (Table 2).

##### Unit Bias, Package Size, and Dishware

Most mothers discussed using package size as a cue for an appropriate portion size to serve (*“It’s generally based on the packaging I think. It does influence you. So if we are out and about and there’s a packet of crisps or biscuits or a smoothie or a yoghurt, I’ll just think yeah that’s fine. At home, I think you have more control don’t you, so you can put it in a bowl” P35, son, 39 months*). Package size acted as both the minimum and maximum amount mothers would offer to their child at any one time. Therefore, when children requested additional servings, mothers found it easier to say no and communicate portion size limits with their child (*“They’re actually quite helpful (packaged snacks). I can say that is your snack, you can eat what’s in there but then there is no more. I think for them as well they understand a bit more when they get to the bottom of the packet, they have all gone and that’s it” P32, son, 31 months*). Other mothers mentioned removing snacks from their original packaging and serving them on plates/bowls. Dishware size acted as a cue for mothers to determine how much to serve to their child independent of the type of food on offer (*“See we’ve actually got a small plastic bowl that I would normally serve her from, so I use those as a way of judging things. It’s funny actually I don’t even think about it, I get the same bowl every time and I just look at what it looks like in the bowl and use that as a judgement” P27, daughter, 45 months*). Some mothers preferred using dishware to packaging, since they could visualize the quantity served to the child and be in full control of how much their child receives. This method was used for all food types, e.g., LED and HED snacks. (*“The mini breadsticks. They come in quite a big bag, far more than I would give for a snack. If he ate them all he wouldn’t eat his meals so we will open a pack and put some in his little snack pots and then seal them up” P36, son, 33 months*). Alternatively, mothers referred to using their or their child’s hand size to determine a suitable snack portion size to serve (“I usually do my handful and then a couple more” P22, son, 29 months) (*“I use the palm of your hand thing; obviously she has a smaller hand, so I might have a whole apple so she will have a smaller apple” P15, daughter, 45 months*).

##### Sharing Snacks

Mothers discussed sharing snacks between multiple children or themselves to ensure their child received a reduced portion size (*“If it was say a biscuit, I might kind of share one with her” P27, daughter, 45 months) (“Well it would usually be him and his brother so I would probably do this *breaks in half* and give half to him and half to his brother” (chocolate coated cookie), P25, son, 35 months*).

##### Subdividing Larger Portions

During the think-aloud task, mothers subdivided large portion sizes into smaller units before serving. This included chopping (grapes) or breaking an original larger unit (chocolate-coated cookies) into one or multiple smaller units. When asked about this, mothers said this was a method used to make their child feel like he/she was receiving a larger quantity of food (*“We would cut these up for him obviously so that it looks like slightly more for him” P13, son, 25 months*).

Alternatively, mothers discussed setting minimum and maximum portion sizes that they would be happy to serve to their child at one snack occasion. Some would then choose to provide the minimum portion size from a larger serving in the first instance knowing that their child would request more. Often the child would receive a second serving but mothers ensured the total snack portion size would not exceed their perceived maximum portion size (*“I’d probably start with not that many crisps because she would probably ask for more. So, I’d probably go for a little handful but assume she would probably have some more” P27, daughter, 45 months*).

##### Unthinking, Automatic Processes

Despite the variety of portion control methods discussed/observed during the think-aloud task and interviews, most mothers were unable to verbalize portion control methods used or give reason for the portion size that they served (*“I don’t really think about it, I just kind of do it without thinking really” P9, son, 47 months*).

#### 3.2.3. Theme 3: Awareness and Use of Portion Size Recommendations

Mothers reported confusion about portion size recommendations for preschool children. They discussed the nutritional information they were aware of and the sources of these. Barriers to following recommendations, as well as which agencies to trust for portion size guidance, were also discussed. The importance of ensuring information is from a trusted source, easily accessible, and clear was mentioned (Table 2).

##### Confusion around Portion Size Guidance for Snack Foods

Mothers mentioned receiving information regarding the types of food they should be offering to their young children and were aware of/or had used portion size guidelines for adults. However, most mothers were unaware, or simply did not know, if portion size guidelines for preschool children exist and believed that many other mothers felt this way (*“I’m sure that there are some (portion size guidelines) actually, no, I’m not. I don’t know what they are” P1, son, 42 months*). Despite mothers being unaware of portion size recommendations for their children, when thinking about it, they presumed they were probably providing their children larger than recommended amounts and that packaged snacks are too large for preschool children. For those mothers aware of portion size recommendations, they felt that, in some cases, portion size recommendations of HED and LED foods are too small (*“I have occasionally seen a thing that says they should have like three chips or something and I’m like that’s not going to work for my child” P8, son, 29 months*) (*“I did look it up on the internet (portion size of broccoli) and I was really surprised how small it was actually” P16, son, 48 months*). Furthermore, mothers thought portion size recommendations for preschool children are not easily accessible or well-advertised.

##### Trust/Mistrust of Sources

Mothers reported that their primary source of information about solid food introduction came from a health visitor (a trained nurse who specialises in maternal and child health). They reported that they received information on complementary feeding but no portion size information was given. Mothers mentioned using online sources and social media groups to gain information, but again this information was focused on complementary feeding (*“I remember years ago when you wean, you get a health visitor but I don’t remember talking about portion sizes, I don’t recall that” P14, son, 47 months*).

Mothers felt that adhering to recommendations would be difficult when their child was in the care of others (fathers and grandparents), as mothers perceived the child’s father to provide larger portion sizes than themselves. However, despite this barrier, mothers expressed a desire to see portion size guidelines more readily available for preschool children. They emphasized that guidelines must be clear, child-centered, realistic, and from a trusted source. There was no preferred format for the information other than it being clear and easily available (*“I do think guidelines, they need to write them in an easy to understand way so you can maybe pin it to the fridge and it be simple and it would be easy” P18, daughter, 28 months*). Examples were provided such as online, in leaflet format, or via a health professional. Mothers also felt that the government and food industry should take responsibility to educate families about appropriate portion sizes. (*“I don’t really trust the portion sizes that are out there anyway. They are made by manufacturers. If the government thinks there’s a problem with kids getting too much, eating too much rubbish, then I think they do have a responsibility to at least educate the public” P14, son, 47 months*).

##### Importance of Packaging as a Guide to Portion Size

Mothers tended to rely on packaging as an indicator of what constitutes an appropriate portion size. Furthermore, mothers highlighted that packaged snacks are convenient and cost-efficient, they guarantee consumption, and they are cheaper than fresh fruit (*“The reason I like the little bags is they are handy and you can take them out and about” P24, son, 24 months*).

### 3.3. Quantitative Results

#### 3.3.1. Portion Size Served Versus Recommended Portion Size

Table 3 shows the means and standard deviations of snack foods mothers prepared for themselves or would serve their child at an afternoon snack time. Four out of five snack foods served to children were significantly different to recommended amounts. Mothers served their children larger than recommended amounts, as determined by More and Emmett [17], of salted potato chips (mean difference = 5 ± 7 g, 28 kcal; *p* < 0.001), chocolate-coated cookies (mean difference = 6 ± 8 g, 30 kcal; *p* < 0.001), and white grapes (mean difference = 26 ± 33 g, 18 kcal; *p* < 0.001), and smaller than recommended amounts of cereal (mean difference = 4 ± 5 g, 15 kcal; *p* < 0.001). Similarly, mothers served themselves significantly more grapes than recommended (mean difference = 24 ± 46 g, 17 kcal; *p* < 0.01). All other food items (salted potato crisps, chocolate-coated cookies, carrot, and cereal) were served in line with recommended amounts (*p* > 0.05).

#### 3.3.2. Correlation between Maternal and Child Portion Size

Pearson’s correlation was conducted to explore the association between maternal and child portion size selection. The analysis revealed a positive and significant correlation between maternal and child portion size selection of crisps (*r* = 0.63, *p* < 0.001), carrot (*r* = 0.43, *p* <0.01), and cereal (*r* = 0.59, *p* < 0.001). There was no significant association between maternal and child portion size of grapes (*r* = 0.24, *p* = 0.13) or cookies (*r* = 0.31, *p* = 0.056).

## 4. Discussion

The primary aim of the present study was to explore what factors influence mothers’ decisions and judgements about a suitable snack portion size to serve preschool children and to explore what portion control methods are used in the home environment, using the think-aloud method and semi-structured interviews. The results demonstrated that decisions regarding snack portion sizes are complex, dynamic, and centered around three main themes: portion size considerations, portion control methods, and awareness of portion size recommendations. Mothers adjust the portion sizes that they serve based on personal feelings of hunger, how much their child consumed so far and proximity to the next meal, their children’s behavior or appetite, and the perceived healthiness of the food item. Food packaging often acts as a minimum and maximum portion size to serve with other portion control methods including the use of bowl size, hand size, or sharing food between family members. Mothers reported confusion about recommendations for snack sizes and called for such information to be accessible, child-centered, and simple. The secondary aim of the study was to compare snack portion sizes served in the home environment to recommended amounts. Caregivers self-served portion sizes tended to reflect recommended portion sizes for adults. However, four out of the five snacks foods served to children were significantly different to portion size recommendations for children aged 2–4 years [17]. As previously reported [15], a positive correlation was observed between maternal self-selected portion size and child portion for most snack foods, except grapes and chocolate-coated cookies.

Factors within the external environment influenced the snack portion size mothers served to their child at an afternoon snack time. For example, snacks offered in close proximity to a meal tended to be smaller than their usual offering, or, if their child missed a meal, then a larger snack portion size would be offered. These findings support those reported in a qualitative study exploring low-income mothers’ perceptions and use of portion size strategies in the USA, whereby mothers were less likely to offer a snack in close proximity to a meal [13]. In the present study, maternal personal feelings of hunger were also found to influence portion sizes served to children, suggesting that mothers may transfer their personal hunger onto their child, regardless of their child’s actual hunger and energy needs. Similar findings were demonstrated in American mother–child dyads at a buffet-style meal, whereby mothers who were hungry perceived their child to be hungry and, thus, served their child a larger meal [16]. As such, educational interventions focusing on techniques to evaluate preschool children’s hunger, independent of personal feelings of hunger, may be beneficial.

Mothers were also influenced by the perceived healthfulness of a food, which determined whether portion size restrictions were enforced and to what degree. These beliefs support previous research, whereby mothers reported that providing children with a balanced diet was of greater importance than providing appropriate portion sizes [31,32]. In the present study, “healthier snack foods”, described as foods low in salt and free sugars (i.e., grapes), were served in larger than recommended portion sizes for both children [17] and adults. Furthermore, mothers discussed providing their child with unrestricted access to fruit and vegetables as a method to encourage consumption. Providing fruits and vegetables as snacks to young children might confer an advantage compared to offering them as part of a meal, since offering fruit and vegetables in the absence of competing foods is associated with increased consumption [33,34]. For example, in a home-based study, mothers were randomly allocated to reduce the portion size of their child’s HED snacks by 50% or replace HED snacks with fresh fruits and vegetables. When fresh fruit and vegetable snacks were served in isolation of HED foods, children consumed significantly more vegetables and reduced their total daily energy intake [35].

Mothers discussed restricting portion sizes of HED foods by breaking items in half or eating part of a packaged snack so their child received less. Blake et al. [13] reported that mothers perceived HED snacks to be less healthy due to their high sugar and fat content, resulting in mothers offering their child a reduced or restricted portion size. Previous research demonstrated that parents restrict HED snacks/foods in order to prevent the development of overweight and obesity in their children [31] and to prevent poor dental hygiene [14]. However, restricting access to HED foods can result in adverse eating outcomes including increased desire for and consumption of a food once the restriction is removed [36]. Energy regulation in the short term was thought to be accurate in infants and young children [37,38], but recent evidence shows that even babies under 12 months fail to compensate for energy and the ability worsens with age [39]. Therefore, it might be beneficial to reduce the portion size of HED snacks offered to children and offer LED foods such as fruit and vegetables [35].

Based on past feeding experiences, some mothers were confident that they learned how much their child would usually consume and were able to adjust portion sizes based on features of the child such as food liking and behavior. For example, in some instances, liked foods were reported to be offered in large portion sizes as per child request to avoid creating upset, a finding that was previously reported by a cohort of UK mothers of children aged 3–5 years [14]. In Carnell’s study [14], mothers discussed emotional feeding practices, whereby they provided chocolate or crisps to control their child’s behavior or to prevent upset. In contrast, disliked items were offered in reduced portion sizes, or not at all, to prevent food waste and associated financial costs. These findings align with those reported by Johnson et al. [10], who indicated that low-income African American and Hispanic mothers are financially constrained by food waste and have a strong desire to avoid wasting both time and money.

Within the home environment, a variety of portion control methods were utilized, consistent with methods described previously by a cohort of low-income mothers in the USA [13] and mothers attending a Head Start center in southwestern USA [40]. For example, mothers demonstrated subdividing large portion sizes into small containers, breaking items into smaller pieces, or sharing snacks between family members. Mothers also mentioned using a bowl or spoons to measure portion sizes; however, these vary widely in their size, thus emphasizing mothers’ uncertainty regarding appropriate portion sizes [40]. In the current obesogenic environment, family/share size foods are easily accessible and of good value for money, which makes these items more appealing to the buyer. However, simple portion control aids such as scoops, cups, or hand sizes shown on packaging may be useful to parents in promoting appropriate portions for children.

Mothers expressed a preference for serving their children pre-packaged snacks since they are well liked and convenient, and they provide a portion size limit that can be communicated to children. These findings are consistent with previous research in the USA, where mothers reported reliance on pre-portioned snacks to simplify, or replace entirely, their decision on the portion size to serve [13]. Pre-portioned snacks are typically larger than age-appropriate for young children [3] and may explain the finding reported here of mothers serving their children snacks in portion sizes larger than recommended. One solution might be for the food industry to increase the availability of smaller packaged snacks or to offer more nutritious options [13]. However, this may require industrial modifications which may not be environmentally friendly or of sufficient profit to the food industry. Instead, it may be more appropriate to encourage feasible methods of downsizing in the home environment such as snack reduction or replacement with fresh fruits and vegetables [35].

Awareness (or lack of) and use of portion size recommendations for children became a prominent discussion point in interviews. Consistent with previous findings [41], mothers were confused and unaware about the existence of portion size recommendations for snack foods. In any case, the UK Eatwell Guide and other similar resources (e.g., reference [42]) simply state that “treat” foods should be eaten less often and in small amounts with no further indication as to what constitutes a “small” or “child” portion. Instead, mothers referred to following advice from health visitors which focused more on complementary feeding and types of food to offer rather than how much to offer. Moreover, previous experiences of adhering to guidelines for adults were discussed, demonstrating mothers’ competence and willingness to follow advice. In previous studies, mothers generally expressed an unwillingness to weigh foods [31,43] and, thus, advising parents to weigh foods might not be the best approach. However, simple visual guidelines or measures such as those proposed in the British Nutrition foundation “Find Your Balance” [44], which recommends the use of handfuls, etc. as portion size measures, might be more effective than weighing.

In the present study, following portion size advice when their child is in the care of someone else (e.g., father or grandparents) was identified as a barrier to portion control. Informal care providers such as grandparents, friends, and babysitters are always an important source of childcare worldwide due to the expanding female workforce and cost of nursery/daycare center [45]. In particular, grandparents are an important source of support in the UK, with over one-quarter of children < 5 years of age receiving care from grandparents [46]. Evidence suggests that fathers and grandparents offer larger portion sizes of less healthy foods to their children than mothers [12,47,48]; therefore, further investigation into the factors that influence fathers’ and grandparents’ portion size decisions may be beneficial for the development of tailored interventions.

In the current study, mothers poured appropriate portion sizes of the snack foods for themselves. However, while mothers “downsized” snacks by offering smaller amounts compared to their self-served amounts, these were still larger than age-appropriate snack portion sizes for their children. Snacks served in larger than recommended amounts for children tended to be energy-dense or contain large amounts of sugar. In contrast, cereals were served in smaller than recommended amounts which might be due to this being a less familiar mid-afternoon snack compared to the rest of the snack foods on offer. Whilst we did not specifically measure liking of cereals, no parent or child reported disliking the cornflakes. Alternatively, this may be attributable to caregiver confusion and lack of awareness of children’s portion size recommendations, consistent with previous work [41], or may reflect child-centered responses such as child liking. Furthermore, the results of the current study suggest a positive relationship between the portion size a mother serves for herself and her child. These results are similar to those reported by Johnson et al. [15], whereby a positive association between maternal and child portion sizes served at meals was observed. Interestingly in the current study, maternal and child portion sizes of grapes and chocolate-coated cookies were not associated. It is possible that this outcome is due to the small array of snacks on offer or due to demand characteristics, whereby mothers did not serve themselves their “usual” portion size because of the presence of the researcher [49].

### Strengths and Limitations

The current study was designed to investigate decisions and portion control strategies employed by mothers when determining the amount to serve their child of a variety of snack foods. This is the first UK-based study to characterize the influences mothers report on amounts they served their child for an afternoon snack. Data were collected in a naturalistic environment to enhance ecological validity. Furthermore, the think-aloud method was used to elicit real-time decisions, reveal portion control methods that were not verbalized in interviews, and rely less on memory. Whilst participants were recruited from varying socioeconomic backgrounds, very few caregivers were of the lowest-income category or from the most deprived neighborhoods. Furthermore, most were white British and relatively well educated despite their place of residence and income. Consequently, the generalizability of findings may be reduced since decisional processes and desire for portion size advice were found to differ between those of middle and high income and education [50], especially in relation to food waste [10]. Future research would benefit from recruiting participants from a more varied background to obtain a greater insight into food portioning practices. Furthermore, mothers voluntarily expressed an interest in participating in the study; therefore, it is possible that they had a prior interest in the topic and may be more health-conscious than other cohorts of mothers, thus further reducing the generalizability of findings. The snacks included in this study were selected based on familiarity and frequency of consumption at the population level [25]; however, the choice of snacks was limited to five, and they were preselected by researchers. Mothers’ and children’s diets are likely to be much more diverse and contain a greater range of snacks than that offered in the current study, and this requires further consideration. It would be insightful to replicate the current study using snacks selected by participants and to include a greater number of snacks. Additionally, we did not collect information on chronic conditions of caregivers, which might have influenced the types and amounts of foods offered to their child. Furthermore, the study was conducted in a naturalistic setting to encourage habitual behaviors; however, the presence of a researcher may have produced a social desirability effect [49]. Quantitative data were collected and analyzed from a limited sample of 40 participants, which might have provided limited power and affected the results. Therefore, these results should be interpreted with some caution. Future studies should seek to focus only on the quantitative aspect of this study and recruit a greater number of participants. Participants’ portion size selection for themselves and their child was compared against portion size recommendations from manufacturer information (based on a 200 kcal/day diet) [20] and those suggested by More and Emmett [17]. In the current study, we did not account for the energy demands of the caregiver or the child; thus, this needs to be given some consideration in future research. Similarly, it is also important to acknowledge that recommended portion sizes communicated to the public do vary [50]. Variability in the accuracy of estimated portion size could be affected by participants’ usual source of portion size information. However, our work demonstrates that portion size selection is complex and depends on a wide range of dynamic factors and not just one single source of information.

## 5. Conclusions

When determining an appropriate snack portion size to serve, mothers were influenced by a variety of factors within their immediate environment, such as maternal hunger, perceived or inferred child hunger, child liking, and perceived food healthiness. Mothers demonstrated the strategies they used to limit children’s portion sizes of certain foods by subdividing large portion sizes into smaller containers or sharing snack foods between family members. The most convenient portion control method was package or dishware size, which acted as both the minimum and maximum quantity to serve. Despite confusion about the recommended portion sizes for preschool children, particularly of snack foods, mothers reported a desire for portion size guidance which is clear, child-centered, realistic, and from a trusted source. The current research demonstrates that decision-making focusing on children’s portions sizes is complex, dynamic, and multifaceted. Establishing a good understanding of the factors that influence the decision-making process can assist in the development of downsizing interventions, for effective communication of portion size recommendations to families with young children.

## Figures and Tables

**Table 1 nutrients-11-03009-t001:** Nutritional value of each snack item (per 100-g serving).

	Energy (kcal/g)	Protein (g)	Total Fat (g)	Saturated Fat (g)	Carbohydrate (g)	Sugar (g)	Salt (g)
Carrot ^a^	0.4	0.6	0.3	0.1	7.9	7.4	<0.01
White Grapes ^a^	0.7	0.4	0.1	<0.01	15.4	15.4	<0.001
Cereal ^b^ (Cornflakes, Kellogg’s ™, ^®^, ©)	3.8	7.0	0.9	0.2	84.0	8.0	1.1
Chocolate-coated cookie ^b^ (Digestives, McVitie’s ^®^)	5.0	6.7	23.6	12.4	62.2	29.5	1.0
Salted potato chips ^b^ (Walkers ©)	5.3	6.1	31.9	2.6	51.5	0.4	1.4

^a^ Low energy density (LED) and ^b^ high energy density (HED), as defined by Albar et al. [8].

**Table 2 nutrients-11-03009-t002:** Quotes supporting the themes constructed from interviews and the think-aloud task.

Theme	Subtheme	Supporting Quotations
1. Portion size considerations	1.1 Features of the environment	“Erm, depends what she’s previously eaten in the day and if I know we are going to have an early tea or a late teatime, something like that. Like today she didn’t eat much at lunchtime so I probably would tend to give her a bigger snack” (P19, daughter, 43 months). “If we are going to do something like swimming, then I might try and make sure she eats more because I know that she needs a bit more energy. Or if we’ve been out in the park, I might give her a bigger snack because I think, well I’d be hungry if I’d been running round” (P12, daughter, 38 months).
	1.2 Maternal hunger and expectations of intake	“It’s the wrong thing to do I suppose but I think how much do I eat, and judge it on that” (P40, son, 31 months). “I don’t know. I think it must be to do with how hungry I am because I think that’s the only way you can really imagine it” (P12, daughter, 38 months). “I do have it in my head that I want her to have had a certain amount in the day. It’s just what I think is an appropriate amount for her age. I have nothing really to gage that against, it’s just when I look at it I think that looks alright” (P3, daughter, 45 months). “More often I think I just judge based on maybe what my parents would have given me as a child or what I see other children having” (P8, son, 29 months).
	1.3 Features of the food	“Generally if it’s healthy I’ll give her lots and lots. If its healthy stuff she can have as much as she likes” (P18, daughter, 28 months). “I think eating too many crisps would be bad for her. I worry about the salt. I worry about fat” (P27, daughter, 45 months). “Carrots, I think they are quite hard to eat so I don’t think I’d give her loads” (P3, daughter, 45 months).
	1.4 Child trait and state food preferences and hunger	“Yeah, it’s very led by him. So, I’m not very good at being boundaried with him so it would be very much, he would choose what he wants and then yeah, that’s how we go about it” (P20, son, 41 months). “He would eat chocolate until it came out of his ears, but obviously he can’t so I do try to limit chocolate and things like that” (P2, son, 40 months). “If I give him a snack and he doesn’t like it, I’m probably not going to give it to him again, because I don’t see the point. There are other snacks available” (P17, son, 41 months). “If I wanted to keep her quiet to get through a more tricky time then I am more likely to give her more. So sometimes I might give more just to keep children quiet” (P3, daughter, 45 months).
2. Methods used to control portion sizes served	2.1 Unit bias, package size, and dishware	“They’re actually quite helpful (packaged snacks). I can say that is your snack, you can eat what’s in there but then there is no more. I think for them as well they understand a bit more when they get to the bottom of the packet, they have all gone and that’s it” (P32, son, 31 months). “Probably if it’s in a packet, yeah, I give the packet. And I think sometimes that means you give them more” (P17, son, 41 months). “Probably for ease I give the whole things quite often but it does depend” (P19, daughter, 43 months). “I’d normally give her what’s in a packet really, in a small packet, so I reckon that’s about right (crisps)” (P23, daughter, 47 months). “If it was a packet of crisps, I’d give one. I go from what the manufacturer packs probably without even questioning it. And like one apple, so like base it on unit size” (P17, son, 41 months). “See we’ve actually got a small plastic bowl that I would normally serve her from, so I use those as a way of judging things. It’s funny actually I don’t even think about it, I get the same bowl every time and I just look at what it looks like in the bowl and use that as a judgement” (P27, daughter, 45 months).
	2.2 Sharing snacks	“Crisps, she would usually share a packet with her cousin, so half a bag” (P28, son, 35 months). “I buy snacks from the supermarket, they are quite good portion sizes because they are snacks for kiddies aren’t they; but if not, if it’s a bigger pack I will just share it so he doesn’t eat it all” (P31, son, 27 months).
	2.3 Subdividing larger portions	“She’s a big fan of grapes. Sometimes I cut them in half to make it look like there’s more” (P15, daughter, 42 months). “We would cut these up for him obviously so that it looks like slightly more for him” (P13, son, 25 months). “Like the chocolate biscuits I just give him one but I know in my mind he might ask for another one. If he asks for another one I will let him have two” (P16, son, 48 months). “I’d probably start with not that many crisps because she would probably ask for more. So, I’d probably go for a little handful but assume she would probably have some more” (P27, daughter, 45 months).
	2.4 Unthinking, automatic processes	“I don’t really think about it, I just kind of do it without thinking really” (P9, son, 47 months). “How do I decide how much I want him to consume? Erm, I don’t know. How do I decide?” (P8, male, 29 months).
3. Awareness and use of portion size guidelines	3.1 Confusion around portion size guidance for snack foods	“Just literally gone on my own ideas. In terms of the advice I sought it was about the type of snacks rather than the portion size” (P17, son, 41 months). “I know like what counts as a portion of vegetables. Like those posters that they put up in the GP ^1^ surgery, but that doesn’t say whether it’s for toddlers or adults” (P14, son, 47 months). “I’m sure that there are some (portion size guidelines) actually, no, I’m not. I don’t know what they are” (P1, son, 42 months). “When I first had my son I did read things. I know 10 grapes is a portion, things like that. I know to use your fist as a size” (P39, daughter, 24 months). “I think parents might generally feed their child more than the recommended amount” (P3, daughter, 45 months). “I think most of the time individual packets of things like pom bears ^2^, or the rice cakes, I think they are very generous for a toddler portion and they may be aimed more at primary school kids than toddlers” (P7, son, 42 months). “I did look it up on the internet (portion size of broccoli) and I was really surprised how small it was actually for kids because I thought it might have been a bit bigger” (P16, son, 48 months).
	3.2 Trust/mistrust in sources	“I remember years ago when you wean, you get a health visitor but I don’t remember talking about portion sizes, I don’t recall that” (P14, son, 47 months). “If there were guidelines it would help, it would make life easier, especially if nursery and school follow them. Although I do think guidelines, they need to write them in an easy to understand way so you can maybe pin it to the fridge and it be simple and it would be easy” (P18, daughter, 28 months). “I don’t mind who provided it as long as I know it was a trusted source” (P39, daughter, 24 months). “A leaflet from the government or the health visitors when they come, I think that would be useful. You know when they have their one-year visit and two-year visit, I think that would be quite useful to receive that (guidelines). It might help with the obesity epidemic” (P16, son, 48 months).
	3.3 Importance of packaging as a guide to portion size	“The reason I like the little bags is, they are handy and you can take them out and about” (P24, son, 24 months).

^1^ A GP (general practitioner) is a medically trained doctor working in primary care. ^2^ Pom bears are the name of a brand of salted potato chips.

**Table 3 nutrients-11-03009-t003:** Snack portion sizes (g) served by caregivers for themselves and their child with a comparison to recommended amounts as proposed by More and Emmett [17], the National Health Service [20], and manufacturer information.

	Snack Food	Recommended Portion Size (g)	Portion Size Provided (g)	Portion Size Served in the Home Environment	
				Mean (± SD)	Range (g)
Adult	Carrot ^a^	80	320	79.0 (49.2)	10–320
	White grapes ^a^	80	320	104.3 (46.0) **	40–320
	Cereal ^b^ (Cornflakes, Kellogg’s ™, ^®^, ©)	30	120	25.5 (14.6)	10–67
	Chocolate-coated cookie ^b^ (Digestives, McVitie’s ^®^)	35	140 (*n* = 9 units)	38.3 (20.1)	12–114
	Salted potato chips ^b^ (Walkers ©)	25	100	27.3 (14.6)	9–100
Child	Carrot ^a^	40	160	40.2 (22.1)	8–95
	White grapes ^a^	40	160	66.0 (33.2) ***	8–160
	Cereal ^b^ (Cornflakes, Kellogg’s ™, ^®^, ©)	18	72	13.7 (5.4) ***	3–24
	Chocolate-coated cookie ^b^ (Digestives, McVitie’s ^®^)	15	60 (*n* = 4 units)	21.4 (8.2) ***	8–38
	Salted potato chips ^b^ (Walkers ©)	10	40	15.3 (6.8)***	7–40

* indicates a significant difference to recommended portion size (*p* < 0.05), ** (*p* < 0.01), *** (*p* < 0.001). ^a^ LED and ^b^ HED as defined by Albar et al. [8].

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
