# Peer review of "Maternal Decisions on Portion Size and Portion Control Strategies for Snacks in Preschool Children"

_nutrients, 2019, doi:10.3390/nu11123009_

Round 1

Reviewer 1 Report

General

The paper describes a relevant topic in a world where food is present in abundance: understanding how mothers decide on snack portion and what strategies they use to control the portion. It seems that little research has been done in this area, so this topic is worth investigating. The think-aloud method is relatively novel, so this adds to the originality of the paper. Generally, the paper is well-written and clear. Still, there are some comments and queries with regard to the manuscript, which need to be changed or explained. These points are outlined below.

Main comments

Abstract line 22: disrupted by ‘immediate demands’ --> when talking about young children, this phrasing seems to imply other things. Immediate demands could refer to a child is crying, because it did hurt him/herself, or needs a clean diaper, so parents need to interrupt what they were doing. This is not what the authors mean. Could you use another word here that more closely refers to what your findings are? Methods/ procedures: How long was the total home visit? The interview lasted 20 minutes, but what about the think aloud method during the preparation and the weighing afterwards? Please add this information. Line 168-169: I don’t think it is proper to refer to Table 2 here, as this is not really a result, but a starting point of the study. Anyway, usually no reference to Tables in data analysis part. Why not combining the information in Table 2 in Table 4? It may be nice to see next to each other: recommended portion, provided portion and actual prepared serving. Table 2: It would be nice to indicate (add) the number of cookies for a 60 and 140 gram portion 3.2.1 The categorization that is introduced at the end of this paragraph does not match with the subsequent headings. I do like the concept of maternal attributes/ characteristics, as I think that line 210-212 may be better situated at the next paragraph, as these points are more related to the mother and perceptions of the mother than really to the environment. So, I would suggest changing this. Line 271-280: Please check carefully, it seems that using dishware as cue is described 2x with the same quote. This doesn’t seem correct. Line 301: Approximately how many mothers or for how many snacks were mothers unable to verbalise portion control methods? Could the authors provide an indication? Line 320: Were the portion size recommendations only mentioned for healthy foods? And were only portion recommendations for healthy food thought to be small? Or were both also the case for unhealthy foods. This is now unclear, since the quote talks about healthy only. Table 3 line 349.... justified?? Table 3: Third quote may fit better with 1.2 than 1.1. See previous comment Line 352: please reformulate this sentence: mean portions and SD... mothers prepared for themselves OR would serve ... since the snacks were not actually served. Line 381: I miss the environmental influence of time of day/ time to next meal... Line 383: I am surprised by the term ‘hand size’... maybe I missed it, but to me, it was new and not described in the text. Line 423: What do you propose? If restriction is not desirable, but kids are also not able to regulate their energy intake well, what do you propose? It would be good to include a direction to end this paragraph. Line 489: I don’t understand the last part. Why would child hunger be influential only for cornflakes? Additionally, did you get any suggestions for a dislike of cereals from the study? Line 515: Please mention the 4 headings as used in the results and tables in the conclusion. Now the environment seems to cover all, whereas in results this is one of the influences. In the conclusion, I also miss the ‘dishware’ as method to guide portion.

Minor comments

Line 26: Should it be ‘into’ instead of in to? Line 44: this highlights... Line 47-48: is portion control a verb? Or should it be to control food portions... Line 69: add the --> the resultant Methods 2.4.4: I assume that anthropometrics was done by the visiting researcher? Part 3.1: spaces not completely consistent around the mean and SD values... Line 190: ‘portions’ may be added Line 228: the ‘l’ is missing in healthy Line 314: believed Table 3 2.1: There (first quote), is this word correct?... They are?

Line 470: Might be nice to give an example of these visual guidelines/ measures, or are these the ones you refer to in line 448?

Reviewer 2 Report

Line 84, Inclusion criteria: caregivers who were >=18 years old and....

Did authors explore the general health conditions of the caregivers, such as chronic diseases, which may have influence on the type of foods included in the kids' diet?

Line 88, .. high street voucher and...

I am not sure what the high street voucher is. Is this the same as gift card? Please add a bit of explanation.

Line 90, The study was carried out in the home environment and took place 2.5 hours following lunch.

Line 99, Mothers were instructed to serve themselves and their child a sandwich for lunch at midday and then....

Line 103, Mothers were provided with 5 snack items one at a time, and invited to verbalize their actions and ....

Please clarify the research protocol. Was the interview completed at each participant's own home or in a "home" setting? Was the type of sandwich served at lunch standardized? Were mother and child interviewed at the same time? Were the snack items provided to the participants after the interviewer arrived? Were the snack items provided by the researcher?

Line 129, Once snack item from each snack food group....

Please add more information regarding the snack food groups. Were these food groups determined based on population preference?

Line 131, (low<2.5 kcal/g, high>2.5 kcal/g).

Where would the "=" 2.5 kcal/g reside?

Lines 136-137, ..adult snack portion size were provided in line with manufacturer recommendations based on a 2000 kcal/d diet and...

Please clarify on the portion size selection for adults. In the procedure section, authors indicated that Mothers were also asked to prepare each snack for themselves. Therefore, the portion size for the snack items seemed were self-selected. Additionally, each mother may have different energy requirement. The 2000kcal/day may be a rough estimate. Please discuss the limitation of this assumption.

Lines 181-182, ... and on average just outside healthy weight range (M=25.5±5.4 kg.m2).

Was the weight and height of the mother collected in the study? or Were these measurements self-reported or measured? Please clarify those demographic data of the mother were collected in the study procedure section.

Line 188, 3.2.1 Theme 1 Portion size consideration

All data included in this study were related to portion size considerations. Could authors use a less general term to capture the theme 1?

Line 325, come from a health visitor.

What is a health visitor? Is this person included nurse, doctor, or any health care provider? Is the health visitor the main information source for mothers? Please provide description about their health/nutrition background.

Lines 375-378, The primary aim of the present study was to explore..... and semi-structured interviews.

The aims of the study have already been stated in the lines 73-77. These text can be eliminated.

Line 504, conducted in a diverse cohort of mothers, however very few caregivers were of the lowest income...

Based on the results in lines 180-182, Most mothers were educated with at least high school (95%), employed (85%), and White British (95%). Does this represent the typical British population to be considered "diverse" or atypical?

Reviewer 3 Report

The study "Maternal decisions on portion size and portion control strategies for snacks in preschool children" comprises a very interesting theme and is, in general, well-written and adequately developed. However, some edits would help improving the quality of the manuscript. Specific comments and suggestions can be found below.

Abstract

Please add a comma in the sentence: “Situational influences were important to mothers, which may mean that planning and portion control are disrupted by immediate demands.”

Introduction

Lines 34-35. About the sentence: “This is known as the portion size effect (PSE) which has been found to be robust and reliable for up to 5 days” – I suggest giving more information about this study or changing the sentence to better explain the “cutoff” of 5 days (in this case, the study only evaluated PSE for 5 days, but it does not necessarily seem that >5days would mischaracterize the results).

Methods

About the sentence: “Exclusion criteria included those with food allergies.” – Who are “those”? Children or their mothers? Please make this information clear. “Sample questions relevant to the research questions were devised.” – Please mention that questions are given in table S1. Line 131: Please close the parenthesis and delete the extra comma at “(Low < 2.5 kcal/ g, High > 2.5kcal/g,”. Line 137: Please rewrite “world health organization” with capital letters (“World Health Organization”) Line 160: The sentence is lacking the end point.

Results

Line 182: Please correct the unit of measurement for BMI (kg/m² instead of kg.m²). Line 185: There is no need to put the “n” if authors already presented the percentage. The other percentages in lines 180, 181 and 183 were not followed by their “n”, so please standardize the presentation. In line 80, children’s age is presented in years, so I suggest keeping the same unit when presenting the mean age in line 185. Lines 261-270: There are sentences of the participants’ speech which are not in italic. Lines 278-280 are repeating the same speech presented in lines 273-276. Line 340: Please replace “my” by “by”. In table 3, I think it would be useful to explain to the reader the meaning of specific terms used by the mothers, like “GP” and “pom bears”. Lines 356-360: I don’t think it is necessary to mention “(t(39)=…)”. The mean difference and the p-value are enough. Line 373: A p-value cannot be negative. Please check the result and correct.

Discussion

There are some important limitations of the study that were not mentioned: the limited sample size may have provided a limited power and affected the results of the quantitative analyses. In addition, limiting the snacks to 5 options also needs to be discussed. Children and mothers’ diets are much more diverse than the food options they could choose in this study, so it is possible that their behavior would be different if really having to choose products they are used to.
